# Educational Simulation Program Based on Korean Triage and Acuity Scale

**DOI:** 10.3390/ijerph17239018

**Published:** 2020-12-03

**Authors:** Jae-Hyuk Jang, Sang Suk Kim, Sunghee Kim

**Affiliations:** 1Rapid Response Team, National Health Insurance Service Ilsan Hospital, Goyang 10444, Korea; njaehyuk@nhimc.or.kr; 2Red Cross College of Nursing, Chung-Ang University, Seoul 06974, Korea; kss0530@cau.ac.kr

**Keywords:** simulation, job satisfaction, nurses, care, triage, emergency service

## Abstract

This study aimed to develop and implement an educational simulation program based on the Korean Triage and Acuity Scale (KTAS) for nurses in emergency medical centers who completed the KTAS training, and assess its effects. We examined the educational effects of the program by evaluating clinical decision-making ability, job satisfaction, and customer orientation among the participants, namely 27 nurses in the emergency center of a general hospital. Data were collected from 3 to 24 May 2017, and analyzed using SPSS 22.0. There was a significant difference in nurses’ mean scores on clinical decision-making ability, job satisfaction, and customer orientation before and after the simulation-based education. In other words, after completing the KTAS-based simulation education program, the emergency nurses showed improved clinical decision-making ability, job satisfaction, and customer orientation. Based on the results of this study, it is expected that this educational program can be effectively used for KTAS education, and it was confirmed that simulation-based education is a useful learning method for triage nurses in emergency medical centers.

## 1. Introduction

According to National Emergency Medical Center statistics, emergency medical centers in Korea are being consistently used by more than 10 million people per year, with 10.343.983 visits in 2015 because of the Middle East Respiratory Syndrome outbreak, 10.419.983 in 2014, and 10.186.341 in 2013 [1]. In addition, the use of emergency medical centers is rising due to an increase in the number of disasters and accidents; higher levels of service at emergency medical centers are being requested because of improvements in individuals’ quality of life due to medical technology developments [2]. Moreover, during the outbreak of coronavirus disease 2019 (COVID-19) in Korea in 2020, the preparedness of emergency departments for triaging patients and ensuring the safety of staff has been of utmost importance [3].

However, the reality is that most Korean emergency medical centers are used as waiting rooms for hospitalizations; consequently, the problem of work stagnation has worsened, leading to the inability of the emergency medical center to provide a satisfactory service to their visitors [4]. The increase in non-emergency patients also leads to inefficient management of medical resources; therefore, the need for a severity classification system to determine and categorize patients’ severity for efficient management has been in the limelight [5]. Severity classification involves the systematic classification of patients visiting the emergency medical center by assessing the severity of the health problem, making a judgment about the order of prioritization of care, performing emergency treatment, and allocating cases to appropriate departments [6]. Previous research found that the emphasis on severity classification has sometimes led to disability in patients because of inadequate classification by medical personnel. In the worst case, it may even lead to the death of the patient; therefore, severity classification should be performed with caution [7,8].

Domestic emergency medical centers use different classification systems, namely the Canadian Triage and Acuity Scale (CTAS), the Australasian Triage Scale, the Emergency Severity Index, or the National Emergency Department Information System in its original version or modified [6]. However, these severity classification tools used in Korea are systems developed in and for other countries. Therefore, the application of these severity classification tools may be complicated in the Korean context because of excessive cost and the difficulty of implementation at the pre-hospital stage [6].

The lack of a standardized severity classification tool that is adequate for Korean emergency settings has been the cause of inefficient management of medical resources and overpopulation of emergency medical centers, and has influenced the decreased satisfaction with the service among visiting patients [1]. Consequently, in 2012, the Korean Society of Emergency Medicine and the Ministry of Health and Welfare developed the Korean Triage and Acuity Scale (KTAS) based on the CTAS, which has been implemented since January 2016 to increase the efficiency of the emergency medical system and the satisfaction of visiting patients. Generally, a standardized severity classification tool can be expected to improve the satisfaction of visiting patients and increase the job satisfaction of nurses tasked with severity classification [4]. The improvement in nurses’ job satisfaction, understood as a favorable attitude toward their job, is also expected to improve their customer-oriented attitude, leading to an improvement in the hospital’s image and customer satisfaction [9]. Customer orientation refers to assessing customer needs and aiming to improve customer satisfaction.

The severity classification at emergency medical centers in Korea is mainly performed by nurses, whose decision making is crucial, as it has a significant impact on patients’ prognosis and treatment process [10]. The Korean Society of Emergency Medicine conducts eight hours of training for the standardized KTAS classification, but this is insufficient to demonstrate a quick and accurate response to patients who come to the emergency medical centers with various and urgent conditions.

Through years of prior research, simulation-based education has been found to effectively cover all aspects of clinical competence, including knowledge, patient assessment ability, motivation, and cooperation with others, as well as clinical performance competence [11], confidence, problem-solving ability, communication, critical thinking ability, self-efficacy, and learning satisfaction [12]. It has also been confirmed as a learning method suitable for practicing in a safe environment before coming in contact with patients [11].

The International Nursing Association for Clinical Simulation and Learning (INASCL) provides simulation-based education for nurses. The standardized simulation curriculum promotes learner-centered simulation-based experiences so that nurses’ improved competencies and skills can be demonstrated in clinical practice [13]. Moreover, simulation education, which is a safe method to train nurses, is very efficient at a time when public interest in medical knowledge is increasing [14]. Roussin and Weinstock [15] introduced the SimZones innovation, a system of organization for simulation-based learning; of four areas describing simulations, Zone 2 simulations include acute situational instruction, such as clinical emergencies, Zone 3 simulations involve authentic, native teams of participants and real patient treatment, and in Zone 4 simulations, through debriefing, the transition to real hospital practice is explained.

There is a dearth of research on the use of severity classification tools by emergency nurses as well as on simulation-based education applied to this field. Indeed, there is a need for the continuous development of simulation-based training programs for the improvement and training of nurses’ skills [16]. Therefore, the purpose of this study was to develop a simulation-based education program based on the KTAS, implement it with nurses in emergency medical centers to assess its effects, and provide basic data to develop systematic KTAS education programs in the future.

## 2. Materials and Methods

### 2.1. Design

The development of the KTAS-based simulation training program for emergency medical center nurses followed five steps—analysis, design, development, implementation, and evaluation—based on the Analysis, Design, Development, Implementation, Evaluation (ADDIE) model, which is a representative model of instructional system design [17]. The development process for the simulation-based Korean severity classification training program using standardized patients is shown in Figure 1.

### 2.2. Participants

The participants in this study were nurses working at emergency medical centers who completed the KTAS training, understood the purpose and contents of this study, and voluntarily agreed to participate. A sample of 30 nurses was recruited based on a minimum sample size of 27, which was calculated with an effect size of 0.50, a significance level of 0.50, and a power of 0.80 using G*Power 3.1.9 (Heinrich-Heine-Universität Düsseldorf, Mannheim, Germany) for Windows, considering a 10% drop-out rate. However, of the 30 nurses, 3 were withdrawn, resulting in a final sample of 27 participants.

### 2.3. Data Collection

Data were collected from 3 to 24 May 2017. After receiving approval and confirmation of cooperation from the participating hospital, the surveys were distributed to emergency medical center nurses. Data were collected before and after the implementation of the simulation education program developed in our study.

### 2.4. Instruments

#### 2.4.1. Clinical Decision-Making Ability

To measure clinical decision-making ability, a tool developed by Jenkins [18] and revised by Baek [19] was used in this study. Clinical decision-making ability refers to the process by which a nurse identifies a nursing problem and finds and selects an appropriate approach. In this study, the severity triage manager was involved in the decision-making process for severity triage through patient assessment, which may be influenced by clinical experience, education related to previous emergencies, or simulation experience. The tool consists of 40 items in four sub-domains, each with 10 items. A higher score indicates a higher clinical decision-making ability. Reliability was confirmed by a value of Cronbach’s α of 0.82 at the time of development, and 0.86 in the present study.

#### 2.4.2. Job Satisfaction

To measure job satisfaction, the tool revised by Lee in 2010 [20] was used in our study, which consists of 12 items. A higher score indicates higher job satisfaction. Cronbach’s ⍺ was 0.86 in the original study, and 0.88 in the present study.

#### 2.4.3. Customer Orientation

To measure customer orientation, the tool developed by Saxe and Weitz [21] was used. It comprises a total of six items, and a higher score indicates higher customer orientation. In a previous study, Cronbach’s ⍺ was 0.90 [22], and in the present study, it was 0.89.

### 2.5. Data Analysis

Collected data were analyzed using IBM SPSS Statistics for Windows, version 22.0. General characteristics and comparison of key variables before and after program participation were analyzed using percentages, means, standard deviations, and mean comparisons with paired t-tests. The reliability of the study measures was confirmed using Cronbach’s α.

### 2.6. Ethical Considerations

The study plan was reviewed and approved by the participating institution research ethics boards (IRB File No.: NHIMC 2016-12-008-001). The purpose and methods of this study were explained to the participants, and only those who provided written informed consent were enrolled. It was explained to participants that they could terminate their participation at any time during this study, and that this would not lead to any disadvantages.

## 3. Results

### 3.1. Development of the Simulation Education Program Based on KTAS

The simulation-based KTAS education program in this study was developed in five stages: analysis, design, development, implementation, and evaluation, based on the ADDIE model [17].

#### 3.1.1. Analysis

To select an educational topic for the educational program, group interviews were conducted with the study participants in the conference room of the emergency medical center using the question, “Which patient is the most difficult to judge when applying the KTAS?” In the interviews with a total of 30 nurses, difficulties were reported regarding the classification of patients with heart disease complaining of gastrointestinal symptoms, classification of high-risk patients, determination of the treatment range for trauma patients, and assessment of pain and discomfort in children. Approximately 70% of the study participants (21 nurses) reported difficulties in the classification of heart disease patients who complained of gastrointestinal symptoms; thus, this was selected as the subject of the simulation education program.

#### 3.1.2. Design

The educational objective of the program was to quickly and accurately classify severity using the KTAS and assess the patient using therapeutic communication. The simulation program comprised pre-debriefing, scenario operation with standardized patients, and debriefing. The evaluation of the effects of the simulation education program based on the KTAS was designed as shown in Figure 2.

#### 3.1.3. Development

To develop the scenarios, the KTAS manual was used. The scenario was as follows: an adult male visited the emergency room because of sudden abdominal pain, and said that the pain was in a different location compared to previous times; however, as a result of the classification, medical treatment was not required immediately, so he was asked to wait in the patient waiting room. This case was designed to quickly and accurately classify the patient based on the KTAS, and to assess the patient according to the chief complaint through therapeutic communication. The contents of the scenarios were standardized by one emergency medicine specialist, two clinical nurses with more than 10 years of experience in the emergency medical center, and one KTAS instructor. After the pilot test, the scenarios and the problems of the evaluation checklist were revised and supplemented, and the scenarios was finally confirmed.

#### 3.1.4. Implementation

From 3 to 24 May 2017, a total of five times were run for 40 min each. Four to six nurses were grouped together, and each person took 10 min for pre-debriefing, 10 min for scenario operation, and 20 min for debriefing (Appendix A); thus, the total simulation operation time was 40 min.

#### 3.1.5. Evaluation

Clinical decision-making ability, job satisfaction, and customer orientation were evaluated to check the effectiveness of the simulation education program based on KTAS.

### 3.2. Effect of Simulation Education Program Based on KTAS

#### 3.2.1. General Characteristics of Participants

The final sample consisted of 27 emergency medical center nurses. Of the 27 participants, 14 (51.9%) were in their 20s, and the average age was 33 years. Females comprised the majority with 24 participants (88.9%), and the rest were males (11.1%). As for clinical experience, we used a model based on the Dreyfus model of skill acquisition, which was further revised to fit the Korean clinical environment. Clinical experience was divided into four clinical stages excluding the expert stage: beginner (1 year after entering), advanced beginner (2–3 years), competent (4–6 years), and skilled (7 years or more). According to total clinical experience, 14 nurses (51.8%) were considered competent, and there was no one in the beginner stage. As for work experience in emergency medical centers, 14 nurses (51.8%) were considered competent, and the average experience in emergency medical centers was 71.3 months.

#### 3.2.2. Effectiveness of the simulation education program

The results regarding the effect of the KTAS-based simulation education on clinical decision-making ability are shown in Table 1. In terms of clinical decision-making ability, “Review of goals and values” showed the largest effect, while “Search for information and synchronize by connecting with new information” showed the smallest effect.

All 12 items in the job satisfaction tool showed a significant before-after difference (Table 2). “My hospital has many opportunities for promotion” showed the largest effect, while “Considering the above, I am satisfied with my current job” showed the smallest effect.

All six items in the customer orientation tool showed a significant before-after difference (Table 3). “I try to pay attention to the opinions of my customers” showed the largest effect, while “I try to understand what the customer wants” showed the smallest effect.

## 4. Discussion

The purpose of this study was to develop a simulation education program based on KTAS and implement it with nurses in emergency medical centers to assess its impact. After the simulation education program, the participants showed significant improvement in clinical decision-making ability, job satisfaction, and customer orientation.

### 4.1. Development of the KTAS-based Simulation Education Program

In this study, training needs were analyzed through interviews with nurses who were KTAS graduates, who reported difficulties in classifying patients with high risk, determining the range of treatment for trauma patients, assessing pain and discomfort in children, and classifying patients with heart disease complaining of gastrointestinal symptoms. This is in line with the study by Kim, Kim, and Park [23] on the training demands of intensive care unit nurses, the study by Han [24] on the training demands of university hospital nurses regarding clinical skills, and the analysis of demands by Kim and Kang [16] in their study on the development and assessment of effectiveness of a simulation program for novice nurses in specialized units (emergency room and intensive care unit). Therefore, prior research shows that the training demands of nurses who work in emergency settings are higher for specialized clinical skills and knowledge than general nursing knowledge.

Lecture-based education is effective for knowledge transfer, but there are limitations in improving nursing clinical skills [25]. Therefore, our study aimed to develop a simulation-based training program to actualize continuous learning in an environment similar to clinical settings. Scenarios were developed for the classification of heart disease patients who reported digestive symptoms, reflecting previous training demands. We aimed to develop a scenario to diagnose and classify diseases rather than using scenarios related to direct interventions such as cardiopulmonary resuscitation and circulatory diseases used in prior studies [26,27,28]. The developed simulation-based learning program helped with quick and accurate decision making when classifying severity using the KTAS.

Although simulation-based training programs for nurses have been previously developed in Korea, most were related to infection control training [29], basic life support, or advanced cardiac life support [26,27,28,30]. This study is thought to be meaningful in that it confirmed the educational needs of emergency room nurses and provided simulation-based education on the severity classification of emergency patients to nurses who had completed KTAS education.

Recently, there have been studies on simulation-based learning in different medical professions. For instance, Stocker et al. [31] found an increase in crisis management ability among surgeons, cardiologists, anesthesiologists, and nurses in pediatric intensive care units via simulation-based training. Colacchio et al. [32] found improvements in teamwork after implementing simulation-based training with medical professionals in neonatal intensive care units. As emergency medical centers are places where professionals from a variety of fields work together, it is necessary to reenact complex clinical scenarios to increase the effectiveness of simulation-based education, and conduct research on the use of simulation-based education in various medical fields.

### 4.2. Effectiveness of the KTAS-based Simulation Education Program

The second objective of this study was to test the effectiveness of the simulation-based program by comparing the clinical decision-making ability, job satisfaction, and customer orientation of nurses before and after participation in the program. The results are discussed based on each of the three variables.

First, when examining the effectiveness of the KTAS-based simulation program, we found a significant improvement in clinical decision-making ability. Although this is difficult to compare with previous studies given the lack of research on clinical decision making among emergency medical center nurses in the context of simulation training, our results are in line with those from a study by Kim and Park [33] on the development and effectiveness of a simulation training program for novice nurses in special units (emergency room and critical care unit), and a study by Maxson et al. [34] that found improved clinical decision-making ability after a multidisciplinary implementation of simulation-based training. Furthermore, studies on clinical decision-making ability [19,33,35,36] also reported a significant effect of job satisfaction, critical thinking disposition, and autonomy on clinical decision-making ability. This is similar to our results, as the opportunity to learn through repeated training and on-site reenactment learning in a safe environment may have led to improved clinical decision-making ability. Moreover, simulation-based education involving reenacting clinical situations may be especially useful to new nurses who lack clinical decision-making ability and clinical skills.

Second, after implementing the KTAS-based simulation program, we found a significant increase in job satisfaction, signaling that the simulation was effective. This is similar to the results of Cheng et al. [37], who found increased satisfaction after simulation training for pediatric emergency situations for various medical professionals. The increase in job satisfaction in our study may have been due to increased self-confidence because of the simulation training in a safe environment that is similar to a clinical situation. Based on this, education and training using simulations may help to improve the satisfaction of health professionals and medical staff. In addition, a follow-up study related to this is necessary.

Third, the results of the evaluation of the KTAS-based simulation program showed a significant increase in customer orientation, indicating that the program was also effective in this respect. This is in line with a study on the effect of hospital nurses’ perception of inter-marketing, empowerment, and job satisfaction on their customer orientation [38], a study on the effect of customer orientation, emotional labor, exchange relation, and close ties on the willingness to change jobs in hospital nurses [39], and a study on the effect of specialty professionalism of specialized hospital nurses on job satisfaction, customer orientation, and quality of service [40], these studies found that increased customer orientation was associated with higher job satisfaction, lower willingness to change jobs, and more active inter-marketing.

Despite the increasing demand for high-quality services in emergency medical centers, the satisfaction rate among emergency room visitors is low at 42.1%, with emergency medical costs and waiting time as factors influencing dissatisfaction [1]. Kindness toward medical staff (doctors, nurses) increased slightly compared to 2014 (53.9%) in response to 57.1% of responding that they were satisfied, but it was found that patients who visited the emergency room felt that the quality of service was low in relation to the cost [1]. Based on this, improvement in nurses’ job satisfaction will lead to improved customer orientation, which further leads to an enhancement in the image of the hospital. Thus, the customer orientation improvement seen in our study is a relevant result for the improvement of satisfaction among inpatient and outpatient visitors. Furthermore, in a medical setting where the demand for customer service is increasing, the improvement of customer orientation found in this study is considered a finding that meets the public’s expectations regarding future emergency medical centers.

Overall, our study found that the KTAS-based simulation education program had a positive effect on nurses’ clinical decision-making ability, job satisfaction, and customer orientation. Therefore, simulation-based nursing programs using standardized patients must be developed targeting various clinical skills and treatments required in emergency medical centers. Moreover, using simulation education based on the KTAS can lead to more realistic training, and the resulting ability to quickly and accurately classify severity can have a positive effect in the prognosis and satisfaction of patients.

This study also had certain limitations. First, this study did not include a control group for comparison. Second, additional evaluations were not performed over time to confirm the effects of training in the longer term. Third, the association between general characteristics such as age or clinical experience and the study variables was not examined. Finally, we did not investigate highly crowded emergency medical centers as a potential context; therefore, future studies should develop a scenario related to overpopulation in the emergency room that requires a quick decision on severity classification, such as a scenario where an emergency patient and a non-emergency patient are admitted simultaneously to the emergency medical center.

## 5. Conclusions

To determine the efficacy of simulation-based education, it is necessary to put it into practice. This study confirmed that the KTAS-based simulation education program developed for emergency medical center nurses was effective in improving their clinical decision-making ability, job satisfaction, and customer orientation. The results of this study provide evidence for the application of simulation-based education in the clinical field, deviating from traditional education methods. In addition, this education program was helpful in improving the professionalism of emergency nurses.

## Figures and Tables

**Figure 1 ijerph-17-09018-f001:**
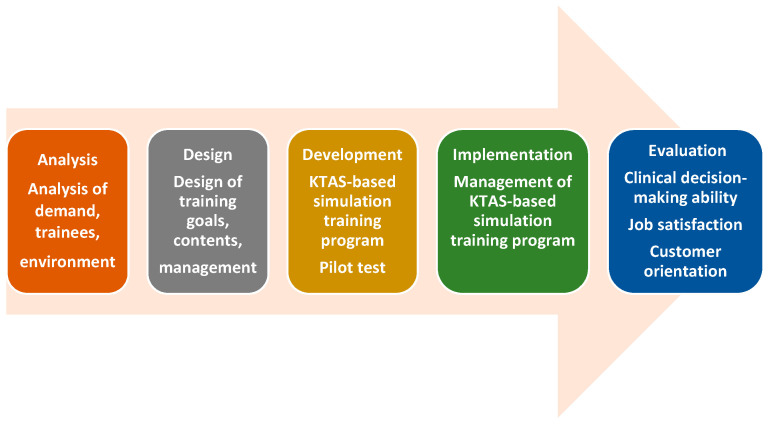
Development of the stimulation training program based on the Korean Triage and Acuity Scale (KTAS).

**Figure 2 ijerph-17-09018-f002:**
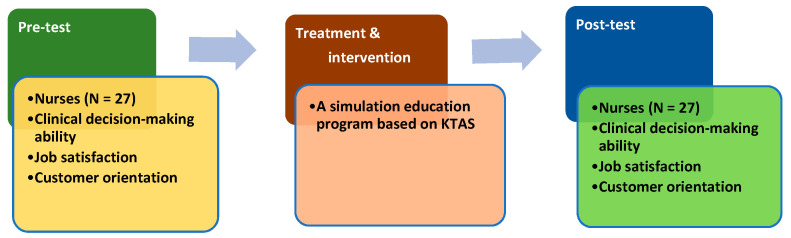
Evaluation of the simulation education program based on the Korean Triage and Acuity Scale (KTAS).

**Table 1 ijerph-17-09018-t001:** Effectiveness of simulation education program: clinical decision-making ability.

Items	Before Participation M ± SD	After Participation M ± SD	*t*	*p*
Clinical decision-making ability	2.66 ± 0.25	3.95 ± 0.36	−27.19	0.000
Evaluate and re-evaluate the conclusion	2.69 ± 0.24	3.93 ± 0.30	−15.40	0.000
Review of goals and values	2.59 ± 0.32	4.06 ± 0.31	−21.33	0.000
Search for information and synchronize by connecting with new information	2.70 ± 0.23	3.95 ± 0.29	−19.26	0.000
Investigate about the choice and the alternative	2.66 ± 0.23	3.92 ± 0.24	−18.08	0.000

*p* < 0.05.

**Table 2 ijerph-17-09018-t002:** Effectiveness of simulation education program: job satisfaction.

Items	Before Participation M ± SD	After Participation M ± SD	*t*	*p*
Job satisfaction	2.65 ± 0.56	3.86 ± 0.47	−16.27	0.000
I can work independently	2.89 ± 0.89	3.81 ± 0.56	−4.65	0.000
There are many opportunities for me to be recognized for my ability	2.59 ± 1.01	3.92 ± 0.61	−6.45	0.000
I can share with my superiors any dissatisfaction or grievances about my duties	2.81 ± 0.96	4.19 ± 0.79	−6.60	0.000
I feel sufficiently rewarded and fulfilled by my job	2.59 ± 1.05	3.89 ± 0.80	−4.69	0.000
My salary is at a satisfactory level	2.93 ± 1.04	4.04 ± 0.75	−4.22	0.000
I am receiving a fair reward for my work	2.74 ± 0.81	3.82 ± 0.73	−5.82	0.000
The amount of work I do is adequate	2.44 ± 1.01	3.63 ± 0.79	−4.53	0.000
My hospital has a good welfare system	2.48 ± 1.22	3.85 ± 0.77	−5.87	0.000
The conditions necessary for performing my duties are well established	2.52 ± 1.12	3.96 ± 0.71	−5.18	0.000
My hospital has many opportunities for promotion	2.41 ± 0.64	3.81 ± 0.83	−6.98	0.000
There is effective personnel management at my hospital	2.67 ± 0.97	3.74 ± 0.59	−5.60	0.000
Considering the above, I am satisfied with my current job	2.74 ± 1.26	3.74 ± 0.90	−3.08	0.005

*p* < 0.05.

**Table 3 ijerph-17-09018-t003:** Effectiveness of simulation education program: customer orientation.

Items	Before Participation M ± SD	After Participation M ± SD	*t*	*p*
Customer orientation	2.67 ± 0.34	3.88 ± 0.69	−11.44	0.000
I respond well to customer questions	2.74 ± 0.67	3.85 ± 0.86	−6.18	0.000
I am kind to my customers	2.56 ± 0.89	3.85 ± 0.77	−5.60	0.000
I try to understand what the customer wants	2.74 ± 1.06	3.81 ± 0.74	−5.04	0.000
I try to help the customer	2.52 ± 0.89	3.81 ± 0.96	−5.75	0.000
I try to pay attention to the opinions of my customers	2.78 ± 0.75	4.00 ± 0.73	−7.50	0.000
I am truly interested in the customer	2.67 ± 0.92	3.96 ± 0.65	−5.75	0.000

*p* < 0.05.

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
