# Peer review of "Educational Simulation Program Based on Korean Triage and Acuity Scale"

_ijerph, 2020, doi:10.3390/ijerph17239018_

Round 1

Reviewer 1 Report

Jang and colleagues developed and implemented a simulation study to enhance training for nurses in utilizing the patient severity classification system. Overall, it is great to see that experimental learning and simulation are being used as they have been proved to be more effective for adult learners. Here are my specific comments:

  • There are many running sentences (e.g., Line 212-217 is just one example). Those sentences must be modified and rewritten to support the message that the authors want to convey
  • The quality of visuals can be improved. Images are blurry and have low quality
  • In figure 2 (page 4) the samples size is shown as “N=27”. However, it is on page 5, that it is discussed three of the disqualified surveys were eliminated.
  • In Results: 3.1.1 Analysis, only one specific case was discussed and it is not explained “how” or “why” heart disease patients category was selected for the simulation study. How about other categories that were identified in needs analysis (lines 191-192) such as “classify patients with high risk”, “determine the range of treatment for trauma patients”, “assess pain and discomfort in children”?
  • On page 5, line 167, it is not explained how the Dreyfus model was modified to fit the Korean clinical environment. What are the specifics of these revisions?
  • The results are statistically significant, but they are collected during a onetime assessment within a short time window (3 weeks). How is learning retention assessed in long term?
  • Limitations are not inclusive of the actual limitations as some of them are listed above

Author Response

Please see the file attached.

Reviewer 2 Report

The title is very long and contains acronyms, it is recommended to reduce words.

Summary content: OK, although the objectives are missing.

Improve the summary by making the objectives clear.

Author Response

Please see the file attached.

Reviewer 3 Report

I would like to ask about :

  • what type of deabreafing have you used?
  • Do you consider decision making to be a technical or non-technical skill?
    • The Decision-Making Abilities is defined as “An outcome of mental processes (cognitive process) leading to the selection of a course of action from among several alternatives” What kind of biases could your study have?
  • The manuscript describe only one limitation but the professional experience? The level of training? The experience in simulation?
  • In conclusión you say “The results of this study have provided evidence for the application of simulation-based education that could be applied in the clinical field, deviating from the traditional education method”. You haven´t compared the experimental group versus a control group
  • the simulation scenarios should be in supplementary material

Specific comments:

  1. Writing

The writing, structure and organization of the manuscript is in accordance with the guidelines.

  1. Title

The title reflects the content studied.

  1. Abstract

The abstract reflects the manuscript

  1. Key Words

KTAS, decision making and ability and customer orientation aren´t Mesh terms.  

  1. Background

The background don´t reflects the state of the art in relation to the study.

The objective of the study is mentioned, as well as the justification for the choice and importance of studying this theme.

  • International Nursing Association for Clinical Simulation and Learning (INASCL).
  • Roussin CJ, Weinstock P. SimZones: An Organizational Innovation for Simulation Programs and Centers. Acad Med. 2017 Aug;92(8):1114-1120.

  1. Methods

The manuscript don´t describe Eligibility criteria for participants of the study and statistical methods

  1. Findings

Point 3.1 should not be in results

The manuscript report numbers of individuals at each stage of study and give reasons for non-participation.

The results shown are simple. How does age or clinical experience influence the variables analysed?

Discussion

Doesn´t make a summarise key results with reference to study objectives

The discussion is extensive and reasoned.

The manuscript should provide more limitations

Application to Pratice

The practical application of this investigation is explained.

  1. References

Important references are missing such as International Nursing Association for Clinical Simulation and Learning (INASCL).

Author Response

Please see the file attached.

Round 2

Reviewer 3 Report

Thank you very much for responding to the questions raised.
This article is better than the previous one.